# NFTs as a Data-Rich Test Bed:
# Conspicuous Consumption and its Determinants

## Abstract

We show that the market for non-fungible tokens (NFTs), much like the luxury fashion market, exhibits conspicuous consumption dynamics: an NFT's value depends substantially on its social meaning as a signal of wealth, taste, and community affiliation. More specifically, we introduce a novel dataset of NFT transaction data combined with embeddings of the corresponding NFT images computed using an off-the-shelf vision transformer architecture. We use our dataset to identify evidence for two phenomena that prior work has identified as the primary determinants of conspicuous consumption: the *bandwagon effect* and the *snob effect*. For each determinant, we identify characteristics of the NFTs themselves and of the communities surrounding them that drive the effect.

## ACM Reference Format:

Anonymous Author(s). 2024. NFTs as a Data-Rich Test Bed: Conspicuous Consumption and its Determinants. In . ACM, New York, NY, USA, 14 pages. https://doi.org/10.1145/nnnnnnn.nnnnnnn

## 1 Introduction

One of the main ways people signal wealth, status, or community affiliation is through ownership and display of "conspicuous goods:" products, like luxury handbags, which are not valued primarily for their function, but for the social meaning they convey. In 1899, Thorstein Veblen first argued that consumer demand for certain goods and services arises from a desire to establish social affiliations and emulate higher social classes and economic groups [27]. Since then, distinct status-seeking consumption patterns have been found in a variety of contexts [3, 7, 8, 12, 18, 26]. Notably, Leibenstein [18] identified two effects that influence the utility derived from luxury goods: the "bandwagon effect," whereby demand for a luxury good increases as more consumers consume it, and the "snob effect," whereby demand decreases as a good becomes widely adopted.

Subsequent work has identified social dynamics that create and reinforce these effects. Vigneron and Johnson [28] found that conforming with aspirational groups and a desire to be fashionable are primary drivers for the bandwagon effect. Han et al. [14] showed that social structures can significantly influence consumer preferences, with higher-income consumers preferring subtler status signals recognizable only within their social circles. Carbajal et al. [5] shows furthermore that this effect might be most prominent in highly socially connected "old money" individuals. In recent years, many aspects of social life have moved online, and so social signaling is increasingly performed via online communities and digital goods. As a result, researchers are beginning to consider these social signals when assessing and pricing digital goods [11, 13, 19].

In this paper, we argue that *non-fungible tokens* (*NFTs*)—a particular type of digital good—are often conspicuous. NFTs are cryptographically-secured records of ownership, allowing digital goods like images and other media files to be certifiably owned, and thus exchanged. *Profile picture* (*PFP*) NFTs, a popular category of NFTs, are associated with images that are intended to serve as online personae on social networks like $\mathbb{X}$ (fka. Twitter), Facebook, and Farcaster;[1] other NFT categories include those conveying ownership of "skins" or items for gaming characters, other digital wearables, or fine art. All of these types of digital goods are commonly used to signal status (flaunting ownership of something rare or expensive) and community affiliation (using aesthetic choices and token-based network connections to adjudicate membership in a group).

NFTs give us access to a much richer dataset than has previously existed for conspicuous goods: the NFTs we examine are recorded on a publicly readable blockchain ledger, making both the associated digital goods themselves (often images) and their transaction data (who bought; who sold; and the transaction price) globally accessible. This contrasts significantly with the traditional literature on conspicuous consumption, in which researchers have been forced to rely mainly on theoretical models and qualitative

---

[1]"PFP" technically stands for "*p*icture *f*or *p*roof," but in colloquial parlance has also been taken to mean "*prof*ile *p*icture."

measures such as surveys about purchase motivations. Researchers have already begun to leverage the data NFTs provide to learn not only about the NFT market itself [22] but also the forces underlying it. For example, Oh et al. [24] have shown that during their primary sales, NFTs act as Veblen goods, a type of conspicuous good driven by the bandwagon effect. Notably, in the primary market where new NFTs are initially "minted," it is relatively straightforward to identify which collections are popular based on their sell-out rate: collections (endogenously) determined to be trendy sell out entirely, while most others see minimal sales.[2]

Our core claim is that NFTs can be used as a data-rich test bed for research into conspicuous consumption, focusing (unlike Oh et al. [24]) on the secondary market in which NFTs are resold. The long-run secondary market is a natural place to study conspicuous consumption, as the primary market often contains many purely speculative buyers. In Section 2, we describe a novel dataset of image-based NFTs that we constructed, including 48,595,074 NFTs organized into 10,963 collections, and the 3,755,256 unique "wallets" that hold them. Our dataset also goes beyond transaction data, including a subset of images for each NFT collection; this allows us to quantify the visual similarity between individual NFTs and between NFT collections using a pre-trained vision transformer network. We argue that NFTs are conspicuous goods by presenting evidence of both of the primary determinants of conspicuous consumption, and furthermore show that the rich data available in the NFT domain can be leveraged to obtain new insights into the dynamics of these determinants. More specifically, in Section 3, we show evidence of the bandwagon effect and identify two key features that drive it within the NFT market: community affiliation and wealth. Then in Section 4, we show evidence of the snob effect and compare the relative power visual distinctiveness and rarity have in driving sale prices. Finally in Section 5, we discuss the broader value of our NFT dataset and some potential future work.[3]

## 2 Dataset

Our dataset contains information on both individual NFTs and NFT collections. Both the collection and NFT data were collected using a combination of the OpenSea[4] and Alchemy NFT[5] APIs.

*Metadata.* Our NFT data spans 10,963 image-based NFT collections, i.e., series of NFTs associated with images that are organized into collections via unifying smart contracts. (These collections are reflected in "collection pages" on OpenSea.) These collections were mapped using the Opensea API, which automatically filters out "spam" collections (such as re-uploads of an existing NFT collection's image assets). In addition, we filter out collections with very large token supplies, such as the collection created by Rarible, which acts as an art exchange and contains over a billion tokens. Our data contains, for each collection, a list of OpenSea summary

transactional data—e.g., current "floor price" (the minimum price at which any NFT in the collection is currently listed for direct sale) as of January 2024, total sales volume, and average historical sale price—as well as some additional non-sale related metadata data for each collection—category (e.g., profile picture, art, gaming, and so forth) and creation date (as recorded by OpenSea). "Profile picture" ("PFP") NFTs comprise the largest category in our dataset, representing 4,221 of the collections, the next largest being "uncategorized," with 3,192, and "art," with 2,365. The remaining 1,185 collections are split amongst smaller categories such as "gaming," "collectibles," and "photography."

At the NFT level, we store the current owner address of each NFT token in each collection as a owner wallet and token ID pair, collected as of January 26, 2024 (e.g., the owner of Bored Ape #9976 would be recorded in our dataset as Ethereum network address 0 x9c7007B750B509dA0c72338de2C2531eD559F4aF). This gives us 48,595,074 wallet–token ID pairs, associated with a total of 3,755,256 unique wallets.

*Image Embeddings.* Much of the analysis in this paper relies on computing image similarity and training models; both of which require meaningful image embeddings. We computed embeddings using a pre-trained vision transformer network, DINOv2 [25], which takes in an image and produces a 384-dimensional real-number embedding. These embeddings have been shown to give state-of-the-art performance at both image retrieval and image clustering tasks with no fine-tuning necessary [25]; a similar architecture has also been used in NFT price prediction [9].

To retrieve the images, we first pulled and saved the image URLs for all NFTs we wished to retrieve using the OpenSea API. Then, we downloaded the images directly from the source URLs and converted them into $300 \times 300$–pixel PNG files.

We also needed an aggregate notion of an embedding for our collection-level data. In each collection, we randomly sampled 50 NFTs and computed the arithmetic mean (or centriod) of their embeddings.[6] We computed centroids for all 10,963 NFT collections in the dataset.

We would like to have been able to conduct image analysis at a more fine-grained level. However, it was infeasible to generate image embeddings for every NFT in our dataset, as this would have required retrieving and storing over 50 million images. Thus, we instead created a more fine-grained dataset over a smaller number of collections by randomly sampling 1,265 collections from the main sample.[7] For each collection in the subsample, we sampled a total of 600 images per collection, and constructed embeddings as described above, along with centroids based on those 600-image groups. Additionally, we added the average sale price of each individual NFT to this dataset, to aid in analysis we conduct in Section 4.

*Rarity Ranks.* Many NFT images are programmatically generated by combining randomly selected visual "traits." Each trait is often associated with a rarity, usually expressed as the proportion of other NFTs in the same collection sharing the trait. (For a visual example of an NFT image's traits and their rarities, see Fig. 1.)

---

[2]Dworczak et al. [10] introduced a framework of "optimal membership design" that covers the design of networks with cross-member externalities, including NFT communities; the externalities studied there can incorporate conspicuous consumption dynamics such as the bandwagon and snob effects we examine.

[3]One such direction is pursued by a second, companion paper that we have also submitted to TheWebConf 2025. This paper investigates an open question in the conspicuous consumption literature: the market impact of introducing new, visually similar goods on previously existing "reference" goods.

[4]https://opensea.io/

[5]https://www.alchemy.com/nft-api

---

[6]For a discussion of the stability of these centroids with varying sample sizes, see Appendix A.

[7]We rejected collections from sampling if they did not have sufficient sales data for the downstream analysis we conduct.

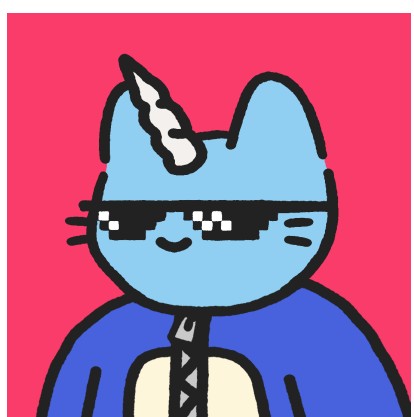

**Figure 1:** *An example of PFP NFT traits.* **Cool Cats #84, image pictured, has the following traits and corresponding rarities:** *Face* **– Sunglasses Pixel (2%),** *Hats* **– Unicorn Horn (1%),** *Shirt* **– Costume Dragon (0.56%). OpenSea (via OpenRarity) ranks Cool Cats #84 at 8,072 out of 9,968 total Cool Cats (one of the least rare NFTs in the collection).**

While assessing the relative rarity of a given NFT seems like a straightforward task, it is made more complicated by the fact that individual tokens—even within a single collection—may have different numbers of traits, and moreover, it is not always clear how to interpret tokens displaying mixtures of both rare and more common traits. Thus, the NFT community has come up with several different ways to measure the aggregate rarity of an NFT. One commonly used rarity metric is a dense ranking computed using OpenRarity[8]; this is the ranking used by OpenSea, and it ranks NFTs first by the number of traits that only appear once in the collection (i.e., "one-of-ones") and then by the information content of the traits. For each of the NFTs in our smaller dataset, we obtained the associated OpenRarity ranks from OpenSea if they were available; of the 1,265 collections, 959 had valid OpenRarity ranks.

*Snob Effect Case Study.* In addition to the 1,265 collections for which we subsampled images, we also gathered more fine-grained data for 9 top sale volume–collections. These collections where chosen by subselecting top volume collections that had a significant correlation with rarity, for more details see Section 4.3. For each of these 9 collections, we computed an image embedding for every image in the collection. We also gathered the entire transaction history for each token in the collection (i.e., a record of every sale, along with the price, buyer, seller, and timestamp).

## 3 The Bandwagon Effect

The *bandwagon effect* occurs when consumers value goods more as they grow in popularity or trendiness, for example because these consumers have a desire for social approval or affiliation with status groups such as the rich and famous [1, 4]. The aspiration to align with the preferences and behaviors of one's community is an important component of the bandwagon effect, and emphasizes the critical role of consumers who are influential within a community

[3]; this aspect of the bandwagon effect is sometimes called the *aspirational effect*, which highlights how community norms and the drive for conformity fuel the desire for conspicuous goods.

In this section, we explore whether NFT values are in part driven by the bandwagon effect. Recent work by Oh et al. [24] showed evidence of a bandwagon effect in the primary-sale NFT market, where popularity is easier to assess because primary sales often follow a bimodal distribution—collections either sell out or have a relatively small number of sales. The secondary market for NFTs presents a different landscape. Here, nearly every NFT being exchanged has already found an owner, making the task of discerning which collections remain socially desirable more nuanced. Furthermore, what is considered popular or trendy can vary across individuals as a function of which other groups they are affiliated with. We model these social dynamics by building a graph with two node sets: wallets and collections, with an edge from a wallet to a collection if the wallet holds an NFT within that collection. This *ownership graph* encodes all of the ownership information in our dataset.

If the bandwagon effect influences a collection's value, then the ownership graph should possess predictive power regarding the value of a collection. Attempting to predict this value using a classical regression model would depend heavily on the features chosen. Thus, we instead use a Graph Neural Network (GNN), training on the ownership graph and asking it to predict the floor price of each of the collections in our dataset. We choose floor price rather than a historical price average because the floor price represents a snapshot of collection value and our graph is a snapshot of ownership data. We also explore how the visual characteristics of an NFT might impact its value. Many collections emulate others in order to take advantage of a trendy aesthetic. To quantify the impact of visual characteristics, we give the GNN access to a collection's centroid as a node feature and measure how this improves performance. Finally, we probe our trained model to get a better understanding of which features it uses to predict value and how they relate to the bandwagon effect.

### 3.1 Measuring Predictive Power

In order to obtain a sensible prediction target, we first transform floor prices into percentiles, sorting each floor price into one of 100 buckets based on which percentile of the distribution (over all floor prices) it lands in. This also gives us a sensible baseline: always predicting the median value of all collections.[9] (We defer the remainder of the experimental setup to Appendix B.)

We compare two versions of our graph neural network (GNN) models against the median baseline. We find that the GNN model without centroids outperforms the baseline root mean squared error (RMSE) of 2779 by 15%, achieving a RMSE of 2356, and the GNN model with centroids outperforms the baseline by 23%, with a RMSE of 2133. Additionally, the predictions of our most effective model, as illustrated in Fig. 2, demonstrate a moderate Pearson correlation with the true values: 0.532 (with $p < 0.05$).

### 3.2 Validating Our Trained Model

The results so far demonstrate that the ownership graph has predictive power, but it is unclear what aspects of the graph drive these

---

[8]https://www.openrarity.dev/

[9]We use median rather than the "50" label because there are point masses in the data.

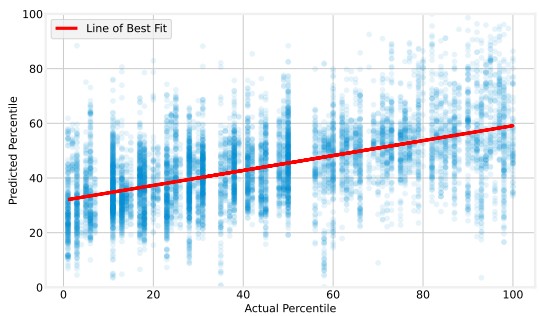

Figure 2: *Performance of the GNN model with centroids.* This graph illustrates the comparison between the true percentile values of NFT floor prices and those predicted by the model. Each point represents an NFT collection, plotted according to its true percentile in floor price ($x$-axis) against the predicted percentile floor price ($y$-axis).

predictions. To address this, we modify the graph—specifically, by adding or removing edges—and observe the impact on the model's predictions. If the model is learning to predict a bandwagon effect, then we should see that adding edges representing ownership of an NFT collection by wallets that are in some sense important or influential should increase the predicted value of that collection; and conversely, removing links to such wallets should lead to a decrease in predicted value.

We define the *importance* of a wallet within the graph as the product of two graph properties: wealth and affinity. We define the *wealth* of a wallet node as equal to the sum of the floor prices of all of its connected collections.[10] We also define a notion of the *affinity* of a wallet node to represent how well its holdings align with the broader NFT-owner community. We compute affinity by first identifying the overlap for each collection, i.e., the number of shared wallets for every collection pair. The affinity of a wallet node is the cumulative overlaps of all its connected collections.

We then modified the ownership graph by first sampling random collections and then, for each collection, sampling non-neighbor wallet nodes to which we added edges, or neighbor wallet nodes from which we deleted edges.[11] We repeated this entire procedure, each time varying the number of edges (25, 50, 100, or 200) to add or delete. In the end, we obtained 50,000 samples for each of the number of edges. We also varied the weights in the sampling procedure—sampling by importance, affinity, or wealth, or uniformly over wallet nodes.

We observed that adding important edges increased predicted percentile floor price on average when compared to the unmodified graph, and this effect became more pronounced as more edges were added. When we added 100 edges, for example, the GNN predicted a higher percentile floor price 99.86% of the time. Figure 3b shows the distribution of changes in predicted percentile floor price when adding edges. Note that when sampling by importance (plotted in

---

[10]This calculation of wealth does not take into account how many NFTs of each collection a wallet owns. This is because our graphs were constructed unweighted and therefore could not have used a more nuanced notion of wealth to form its predictions.
[11]Due to limitations on the number of nodes we can represent in a GPU, each sample iteration is restricted to specific collection node sets.

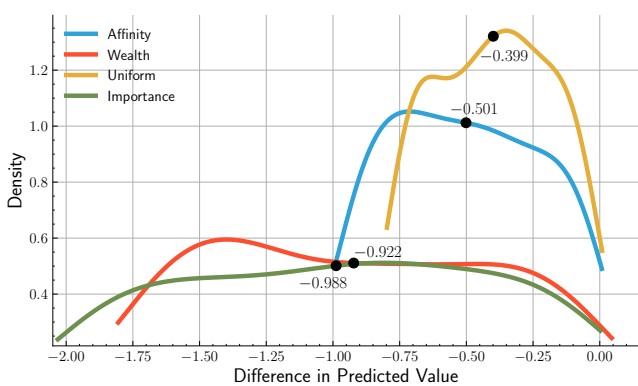

(a) Deleting 100 edges.

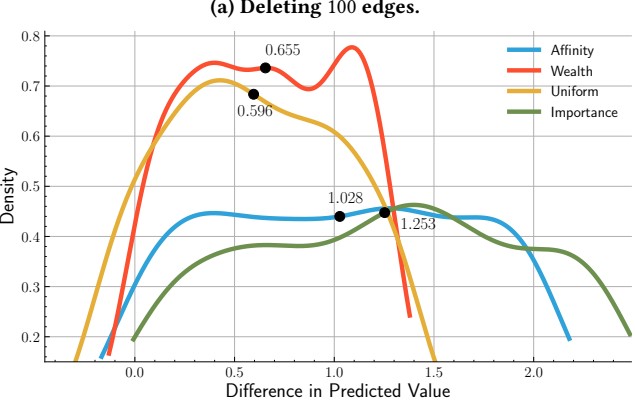

(b) Adding 100 edges.

Figure 3: *Distribution of predicted percentile collection floor price differences on modified graphs, smoothed by kernel density estimation (KDE).* This figure presents KDE plots showing the distribution of differences in predicted values between modified and unmodified graphs. Each line within the plots corresponds to a distinct edge sampling strategy: sampling by affinity (blue), sampling by wealth (red), sampling by importance (green), and uniform sampling (yellow). Means are plotted in black.

green), the floor price the model predicted increased by an average of 1.253 percentiles. We observed similar trends when deleting edges; Fig. 3a shows the associated distribution of changes. These results were robust across the different numbers of edges added and deleted. Results for all sample weightings and numbers of edges added or deleted are presented in Appendix C.

We next looked at the ablation of both affinity and wealth. We observed that when adding edges to wallets sampled by their affinity, the GNN predicted a 56.9% higher floor price on average on the modified graph, relative to when sampling by wealth; moreover, sampling by affinity was associated with higher value predictions overall. However, wealth still provided a meaningful signal, as sampling by importance outperformed affinity. In Fig. 3b, we plot the distribution of predicted differences in floor price when adding edges to wallets sampled by affinity (blue), by wealth (red), and uniformly (yellow) and their means. We also note that as more edges

were added to the graph, the gap between the impact of affinity and wealth grew (see Appendix C); this suggests that adding wallets from tightly connected communities may have a compounding effect on a collection's value.

Conversely, when edges were deleted from the graph, the GNN predicted lower values on average when sampling by wealth than by affinity. Furthermore, unlike when we added edges, the combined sampling approach offered fewer gains; importance changed the model's prediction by only 7.15% more than the next best alternative, as compared to a 22.54% difference when adding edges. This was likely because wallet affinity had less variance when restricted to owners of a specific collection, and was therefore a less informative signal of value. For the full results of predicted floor price movement when deleting edges, see Fig. 3a.

## 4 The Snob Effect

Leibenstein [18] first described the *snob effect* as consumers' values for a good decreasing with its popularity. The concept of the snob effect has come to describe any consumer desire to stand out from a crowd and express individuality through consumption, such as being willing to spend more on goods that are rare, visually distinctive, and easy to show off [7]. The snob effect might seem to be the opposite of the bandwagon effect, but in fact the literature describes the two as operating on different scales: a set of related goods gains average value with popularity (the bandwagon effect) but this same growth in popularity also causes consumers to increasingly value differentiating towards rarer or more distinctive members of this set (the snob effect). Research has also shown that the snob effect can be more pronounced between direct acquaintances [17].

In this section, we explore whether NFT values are in part driven by the snob effect. While we do not have direct access at scale to evidence of explicit social connections between NFT owners, often NFT collections have their own communities with Discord channels, $\mathbb{X}$ accounts, and forums, where holders of the NFTs interact with each other (see, e.g., [16]). Therefore, we expect the snob effect to be particularly powerful among owners of the same collection. We thus explore whether there is a negative correlation between the rarity rank of an NFT and its value within a collection (represented by its average sale price).[12] (We expect the correlation to be negative because the *rarest* NFT in a collection has the *lowest* rarity rank.)

As mentioned in Section 2, rarity ranks are not available for all NFTs. Frequently, these are NFTs without randomly generated traits, making it difficult to calculate an explicit quantitative rarity ranking. We thus rely instead on a quantification of each NFT's *visual distinctiveness*, which we define as the Euclidean distance between its embedding and the centroid embedding of its collection. This distance represents how distinct each NFT is from the "average" NFT in a collection; for example, the images with the smallest and greatest visual distinctiveness within the sample of the Beanz Originals collection in our dataset are pictured in Fig. 4. We aim to determine whether, within an NFT collection, an NFT's visual distinctiveness positively correlates with its average sale price. We also compare the relative power of visual distinctiveness and rarity

---

[12] Negative correlation between sale price and a different notion of rarity has been demonstrated previously by Mekacher et al. [21] on a dataset of 410 collections; however, this is was before rarity rank was easily viewable on marketplaces such as Opensea.

rank as predictors of value because understanding when one is more important than the other may provide insight for conspicuous goods markets beyond NFTs.

Finally, we end the section with a case study of 9 top-sales-volume collections in which we take a deeper look into how these measures impact the sales price and number of transaction for each token in each collection.

### 4.1 Testing the Snob Effect

We begin by examining the relationship between rarity rank and average sale price. We draw these values from our smaller dataset. We calculate the Pearson correlation between rarity rank and average sale price for each collection.

It is worth noting that the NFT market tends to be relatively illiquid at the level of individual NFTs and quite volatile over long time scales. Consequently, two NFTs that most consumers value similarly might nevertheless exhibit significant differences in average historical sale prices, introducing noise into the sales data. Even so, of the NFT collections with rarity ranks available, 67.6% had statistically significant ($p < 0.05$) negative correlation between rarity rank and average sale price. Conversely, only 1.0% collections had significant positive correlation. We repeated this analysis restricted to only PFP collections; since users identify themselves with their profile pictures, we expected a more pronounced snob effect. However, we only observed a small change, with 70.9% of PFP collections showing significant negative correlation between rarity rank and price and 1.0% showing significant positive correlation. The full results can be found in Table 1.

We now examine the correlation between visual distinctiveness and average NFT sale price. We do this analysis both focused on collections that have no obtainable rarity rank as well as on the entire dataset. We compute the Pearson correlation between the visual distinctiveness and average sale price for each collection.

We observed that 24.8% of NFT collections without rarity ranks had significant ($p < 0.05$) positive correlation between visual distinctiveness and average sale price, with only 0.9% collections showing significant negative correlation. When examining all collections in our small dataset including those with rarity ranks, we observed 39.4% showing significant positive correlation and only 0.1% showing significant negative correlation.

Our findings indicate that although visual distinctiveness is associated with average sale price within a collection, its association appears not to be as strong as that of explicit rarity ranks when they are available. One possible explanation is that NFT marketplaces often provide features to easily sort and filter NFTs by rarity, making it straightforward for collectors to assess and trade on rarity, whereas visual distinctiveness is a more subjective and ad hoc measure. We investigated whether rarity ranks always account for more variance in sale prices compared to visual distinctiveness, or if there are collections where the trend is reversed. To determine which factor explains more variance in price, we fit two univariate linear regression models for each of the 959 collections for which we had rarity ranks: predicting price based on rarity rank and based on visual distinctiveness. We then compared the $R$-squared values of the two models for each collection, excluding those where neither model showed a positive $R$-squared. Among the 912 remaining

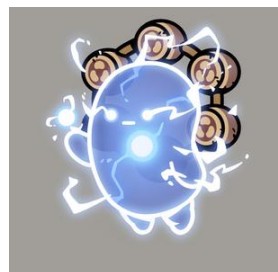

**(b) Bean #9848 – most visually average in our sample.**

**(c) Bean #13956 – most visually distinctive in our sample.**

**(a) Randomly selected Beanz NFT imagery.**

**Figure 4:** *The images with the least and greatest Euclidean distance to the centroid of the images in our small datasets subsample of the Beanz Originals collection.* **Bean #9848 [most average] had an average sale price of** $1.6$ **ETH (**$3,700$ **USD) across the sample period, while Bean #13956 [most distinctive] had an average sale price of** $40.7$ **ETH (**$93,700$ **USD).**

| Predictor | Category | Corr. (+) | Corr. (−) | # Collections | Percent (−/+) |
|---|---|---|---|---|---|
| Rarity Ranks | w/ rarity ranks | 10 | 648 | 959 | 67.5% (−) |
| | PFPs | 7 | 462 | 651 | 70.9% (−) |
| | non-PFPs | 3 | 186 | 308 | 60.4% (−) |
| Visual Distinctiveness | All | 475 | 18 | 1265 | 37.5% (+) |
| | w/ rarity ranks | 392 | 12 | 959 | 40.8% (+) |
| | w/o rarity ranks | 81 | 3 | 326 | 24.8% (+) |
| | PFPs | 313 | 7 | 760 | 41.2% (+) |
| | non-PFPs | 162 | 11 | 505 | 32.1% (+) |

**Table 1:** *Comparison of rarity rank and visual distinctiveness as predictors of average sale price across different NFT categories. This table presents, for each category of NFTs, the number of NFT collections that have significant negative correlation with rarity rank and significant positive correlation with visual distinctiveness (*$p < 0.05$*).*

collections, rarity ranks explained more variance 71.5% of the time, while visual distinctiveness was more predictive 28.5% of the time. Despite the fact that rarity tends to be a better predictor in most cases, there are settings in which visual distinctiveness appears to better explain sale price.

## 4.2 Beyond Linear Correlation

We briefly explored why visual distinctiveness might appear to be a better predictor in some settings. One potential explanation is that rarity ranks may be challenging for linear models. For example, there can be a massive (in particular, nonlinear) price gap between two adjacently ranked NFTs if one of them is a unique one-of-one. We therefore compare the two predictors under Spearman's rank correlation coefficient, which measures monotonic relationships (linear or not).

We computed the Spearman coefficients of the 260 collections where visual distinctiveness explained more variance than rarity ranks in sales data. We then excluded collections without a significant Spearman correlation ($p > 0.05$) for either predictor, narrowing down to 166 collections. Within this subset, 90 collections showed a higher Spearman coefficient for rarity ranks, while 66 demonstrated a greater Spearman coefficient for visual distinctiveness. This suggests that a sizeable portion of the cases where visual distinctiveness explained more variance was likely due to a nonlinear relationship between sale price and rarity.

One rationale for collections whose price is better explained by visual appearance, even in a non-linear model, is that rarity ranks may lose precision in distinguishing the most unique NFTs. It is not uncommon to have a group of NFTs that have a one-of-one (or otherwise especially distinctive) trait, and yet will be sorted by their more common traits when ranked by OpenRarity. In this event, the OpenRarity score may not reliably reflect their "rarity" as perceived by a prospective owner, whereas ranking by visual distinctiveness may provide a more accurate representation. Another explanation is simply that there are collections in which the visual appeal of an NFT image is especially important—for example, if that image is primarily being used as a digital avatar on a social media platform or in an online game. In such cases standing out (or being visually distinct) could have a lot of value.

### 4.3 Case Studies

To further explore the relationship between distance, rarity and NFT sale data we investigated the entire transaction and image dataset we gathered for 9 top collections as described in Section 2. We selected the collections for these case studies by starting with the 30 collections with the highest total sales volume in our dataset and removing those collections that did not have rarity ranks and that our previous study ruled out for not having significant correlations between either sale price and rarity or sale price and visual distance. This left us with 9 collections.[13] The case study dataset has two main advantages: first it contains much more sale price data for every allowing for more statistical power and second it contains additional data on the number of times each token was sold allowing for another dimension of analysis.

We began by replicating the qualitative analysis conducted by Mekacher et al. [21] on our dataset. Mekacher et al. analyzed 3 "exemplary" collections by first binning the rarity of each collection's NFTs into 20 quantiles; they observed that sale price was relatively flat in the lower quantiles but sharply increased in the last (most rare) 2-3 buckets. We saw the same trend as Mekacher et al. [21] in each of the 9 collections that we analyzed; additionally, in cases where visual distance and sale price were meaningfully correlated, we saw a similar relationship, albeit much less pronounced. (For an example, see Fig. 5a.) Mekacher et al. [21] also analyzed the relationship between rarity and number of sales and found a positive relationship. We also observed that the relationship between rarity rank and number of sales appears to have been less driven by outlier values than the relationship between rarity rank and sale price. However, for visual distance, we observed that the relationship with number of sales was usually small or non-existent. (For an example see Fig. 5b; full plots for each of the 9 collections appear in Appendix E.)

We also performed a quantitative analysis of correlations, paralleling our analysis in Section 4.1. In most cases, distance and rarity rank had similar effects on each collection in terms of sale price: typically either both having a negligible effect (Pearson coefficient around 0.05) or both having a more substantial effect (Pearson

---

[13]The initial filtering step actually left us with 10 collections. However, transaction data from the Meebits collection appeared to exhibit substantial amounts of wash trading (self-trading intended to manipulate the price record), so we removed this collection from our analysis.

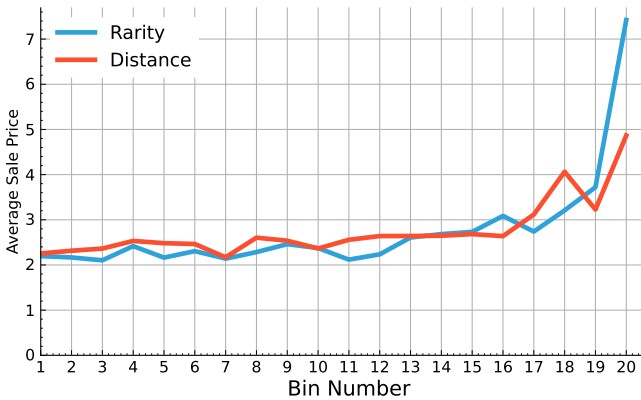

(a) Relationships with Sale Price

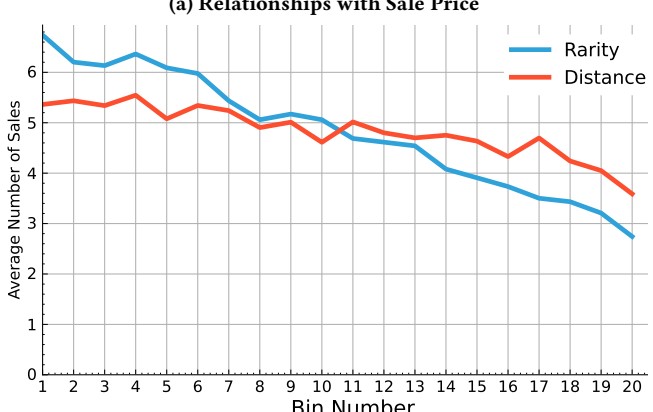

(b) Relationship with Number of Sales

**Figure 5:** *Relationship between quantile bins of rarity (visual distance) and sale price or number of sales respectively in the Cool Cats collection.* **This figure presents rarity and visual distance placed into** 20 **bins by quantiles such that each bin contains** 5% **of the data. These bins are plotted against sale price and number of sales. In the case of rarity ranks bins are sorted from highest rarity rank (least rare) to lowest rarity rank (most rare).**

coefficient greater than 0.1).[14] There are two collections where visual distance and rarity differed in their ability to explain sale price—Azuki and Mutant Ape Yacht Club—yet both of these exceptions in some sense prove the rule. In the case of Azuki, the most visually distinctive NFT images depict the character holding a boombox, which is neither particularly rare nor valuable relative to the rest of the collection. However, they are quite different from the samurai-esque aesthetic of most other images in the collection have. Similarly, in the case of Mutant Ape Yacht Club, the most visually distinctive NFTs depict the apes covered in worms rather than clothes, which is not a particularly valued trait in the community. These results suggest that a more nuanced notion of visual

---

[14]We discuss magnitude rather than *p*-value in this section because we have enough to data for almost every relationship to be statistically significant.

distance, one that allows for multiple clusters within a collection to deal with the existence of multiple common aesthetics, could be important for further analysis.

Next, we quantitatively measured the extent to which how much of the relationship between rarity (visual distinctiveness) and sale price was driven by the most rare or visually distinctive NFTs. To do this, we recomputed correlation with the last 2 buckets (10 percentiles) censored and examined how the correlation coefficient changed. In both cases, rarity and visual distance, we saw a relatively large drop in correlation coefficients with only 1 collection having a Pearson coefficient greater than 0.1 for distance and only 2 collections for rarity. This adds support for the idea that the relationship between sale price and rarity (visual distance) was driven by the most rare (visually distinctive) while the less rare (visually distinctive) almost shared an equivalence class.

Finally, we measured the correlation between rarity (visual distance) and number of sales, and found that rarity tended to be even more correlated with number of sales than with sale price. In the case of Cool Cats, for example, the Pearson coefficient jumped from 0.14 to 0.28. This is potentially unsurprising because the number of sales is a cumulative measure and therefore exhibits much less noise than average sale price, which is confounded by changes in the overall market. However, potentially more surprising is that visual distance tended to be less correlated with number of sales than with sale price. One potential reason for this is that visually distinctive NFTs include both the interesting and more desirable NFTs in a collection but also the "uglier" and more frequently turned over NFTs (e.g., the worm-coated Mutant Apes), creating a lot of variation. For a full table of all of the Pearson coefficients described in this subsection, see Appendix D.

## 5 Conclusion

Leveraging publicly available blockchain data, this paper has argued that the NFT market shows evidence of the determinants of conspicuous consumption and that NFTs are thus conspicuous goods. The richness of this data also allowed us to study the dynamics of a conspicuous market to an unprecedented extent. In our analysis of bandwagon effects, we found that simulating an increase in the number of owners with high affinity for an NFT significantly increased that NFT's predicted value. The predictive strength of affinity in our models suggests that tight community structures may be important drivers of NFT value, a finding that is consistent with anecdotal and ethnographic accounts (see, e.g., [2, 6, 16]). Additionally when analyzing the snob effect, we saw that the publicly visible notion of rarity ranks tended to explain more variance in value than visual dissimilarity, suggesting that signals of exclusivity that are more easily understood and internalized by the market may be especially important for determining value. It will be important for future research to investigate the extent to which these patterns extend to other conspicuous goods markets.

It could be fruitful to consider refinements of the image embedding techniques used in this paper. As mentioned in Section 4.3, some NFT collections contain multiple distinct aesthetics; in these cases, centroids lose information by averaging these aesthetics together. One potential solution would be to allow for multiple clusters within a collection, each with its own centroid; we could

then measure each NFT's distance from their nearest centroid as a more accurate measure of intra-collection distances. Another potentially useful refinement is fine-tuning the vision transformer on NFT images with the task of classifying the collection an NFT belongs to; this might produce image embeddings that more tightly cluster NFTs in the same collection.

Considering topics for future work more broadly, we note that this paper only scratches the surface of potential uses of NFT data for studying conspicuous goods. We take one further step in a companion paper—also under review at TheWebConf 2025—assessing how the values of existing NFT collections are impacted by the introduction of new, visually similar collections. This work is inspired by disagreement in the conspicuous consumption literature about whether knock-offs and look-alikes suppress or raise the prices of luxury goods. There are many additional questions about conspicuous consumption that have so far mainly been studied without access to purchase data. For example, qualitative research has looked at the effect of different types of scarcity on consumers' assessment of the value of conspicuous goods, suggesting that consumers' value assessments may be sensitive to supply-side scarcity ("limited edition") but not sensitive to demand-side scarcity ("almost sold out") [12]. By looking at the impact of collection size (which impacts supply-side scarcity) and market liquidity (which impacts demand-side scarcity) on NFT value, it might be possible to shed light on the magnitude of this effect—although of course there are challenges here because demand-side scarcity may also be a direct proxy for value in steady state. Another question worth investigating is about the existence of "inconspicuous buyers," individuals who only care about signaling to their immediate peers and prefer signals that are hard to decipher by the broader "masses" [14, 20]. "Inconspicuous consumption" is effectively an inversion of the bandwagon effect, that has been thought to occur at the high end of the wealth distribution [14] or even just in well connected "old-money" individuals [5]. Given that we have have access to the full contents of NFT owners' wallets, it may be possible to measure how conspicuous consumption dynamics vary with wealth level, or across different subcommunities of the market.[15]

Despite its overall usefulness as a test bed for studying conspicuous consumption, there do exist important senses in which the NFT market is different from other conspicuous goods markets. Notably, the NFT market is currently patronized by a relatively narrow band of consumers who skew tech-savvy and higher-income (see the discussion in [16]). Conspicuous goods exist across the full range of social strata, and the dynamics may vary across consumer demographics. Furthermore, while data on physical luxury markets is currently hard to come by, a growing wave of NFTs with associated physical-good counterparts (see, e.g., [15, 16, 23]) suggests that it may one day be possible to conduct a version of our analysis for a class of physical goods as well.

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

## A    Embedding Stability

In order to evaluate how stable centroids are across various sub-sampling sizes, we created a dataset containing the full 5000-10000 images for 10 randomly-selected PFP NFT collections.[16] For each of those collections, we computed the "true" centroid that results from averaging all images and compared it to the centroid that is obtained by subsampling a smaller number of images. For each image in our dataset, we then compute distance to both the true centroid and the subsampled centroid, and averaged the absolute values of those differences. We found that even if only 50 images were used to construct the centroid, on average the differences were only 4% of the average distance to the true centroid. In Fig. 6, we plot the average percent difference across various sample sizes.

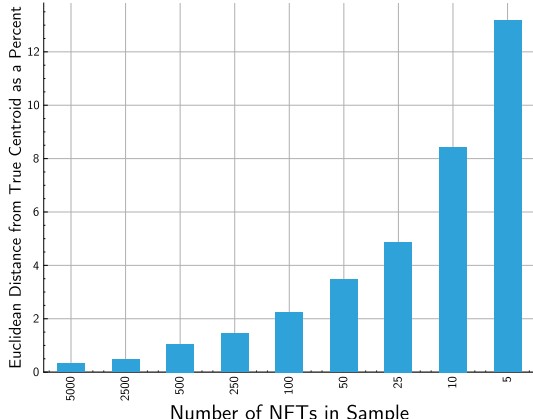

**Figure 6:** *Average percent difference in distance to the "true" centroid and the subsample-constructed centroid.* **For each subsample size** $s$**, we plot the average difference between an NFT's distance to the "true" centroid, computed using every NFT in the dataset, and its distance to the subsample centroid, computed only using a size-**$s$ **random subset of NFTs. The centroids are relatively stable (under** $5\%$ **change in average distance) as long as at least** $25$ **NFTs are used to compute them.**

## B    Bandwagon Effect Experimental Setup

To predict floor price for PFP NFT collections, we utilized a Graph Neural Network (GNN) comprising four Graph Convolutional Network (GCN) layers. The architecture starts with an input layer accepting single-dimensional features (or 384-dimensional features in the case of centroids), progressively transforming these through hidden layers with dimensions of 64, 32, and 16, respectively—with an output layer that predicts a single-dimensional node feature.

This experimental study was conducted on the Compute Canada computing cluster, leveraging both the Narval and Cedar resources. We used 40GB A100 GPUs on Narval and 32GB V100s on Cedar. Due to limitations in GPU memory, we split the ownership graph into 50 subgraphs. The splitting procedure was as follows: We first sampled 75 wallets from the dataset, then we sampled all the collections they held (capping at 1500), and then we pulled all the wallets that held those collections.

Because of our splitting procedure, we needed to be careful in how we split up our training, validation, and test sets. We sampled collection nodes for sets from the global list, and constructed a mapping into the subgraphs. We then trained our GNN with mean-squared error loss on a batch size of 4 subgraphs for 2,500 total epochs. We took the model with the best validation accuracy across those 2.500 epochs which occurred at epoch 1,000.

---

[16]We rejection-sampled to ensure that the collections have at least 5000 images.

## C Supplementary Tables for Section 3

| Number of Edges | Sampling Procedure | Adding Edges | Deleting Edges |
|---|---|---|---|
| 25 | Affinity | 0.494 | −0.265 |
| | Wealth | 0.448 | −0.531 |
| | Importance | 0.575 | −0.612 |
| | Uniform | 0.295 | −0.083 |
| 50 | Affinity | 0.714 | −0.348 |
| | Wealth | 0.523 | −0.746 |
| | Importance | 0.889 | −0.797 |
| | Uniform | 0.468 | −0.336 |
| 100 | Affinity | 1.028 | −0.501 |
| | Wealth | 0.655 | −0.922 |
| | Importance | 1.253 | −0.988 |
| | Uniform | 0.596 | −0.399 |
| 200 | Affinity | 1.481 | −0.614 |
| | Wealth | 0.981 | −1.061 |
| | Importance | 1.523 | −1.257 |
| | Uniform | 0.798 | −0.474 |

Table 2: *Average changes in value predictions.* **This table shows the average change in the prediction of the floor price of all collections by the GNN across each of the graph modification settings.**

## D Supplementary Tables for Section 4.3

| Collection | Sale Price Corr. | # of Sales Corr. | Censored Sale Price Corr. |
|---|---|---|---|
| azuki | 0.083537 | −0.049221 | 0.082964 |
| beanzofficial | 0.126023 | −0.092573 | 0.090254 |
| boredapeyachtclub | 0.066870 | −0.051255 | 0.037238 |
| clonex | 0.130384 | −0.104769 | 0.118309 |
| cool-cats-nft | 0.128013 | −0.115599 | 0.066445 |
| doodles-official | 0.182818 | −0.122661 | 0.075605 |
| mutant-ape-yacht-club | *0.013094* | 0.039136 | *0.003909* |
| proof-moonbirds | 0.183427 | −0.113823 | 0.064193 |
| pudgypenguins | 0.058748 | −0.055114 | *0.004435* |

Table 3: *Pearson Correlations for Visual Distance* **This table shows the Pearson correlation coefficients between visual distance and either average sale price or number of sales for each of the** $9$ **case study collections. The final column shows the relationship between visual distance with the last 10 percentiles censored and sale price. All values are statistically significant (** $p < 0.05$ **) unless italicized.**

| Collection | Sale Price Corr. | # of Sales Corr. | Censored Sale Price Corr. |
|---|---|---|---|
| azuki | −0.170883 | 0.163149 | −0.114551 |
| beanzofficial | −0.165244 | 0.168878 | −0.082399 |
| boredapeyachtclub | −0.063194 | 0.111523 | −0.031122 |
| clonex | −0.183957 | 0.176747 | −0.136650 |
| cool-cats-nft | −0.148992 | 0.281166 | −0.097932 |
| doodles-official | −0.173598 | 0.177977 | −0.045757 |
| mutant-ape-yacht-club | −0.213170 | 0.209414 | −0.141859 |
| proof-moonbirds | −0.204417 | 0.266713 | −0.077283 |
| pudgypenguins | −0.073887 | 0.171108 | −0.020776 |

Table 4: *Pearson Correlations for Rarity ranks* This table shows the Pearson correlation coefficients between rarity ranks and either average sale price or number of sales for each of the 9 case study collections. The final column shows the relationship between rarity ranks with the last 10 percentiles censored and sale price.

# E  Supplementary Figures for Section 4

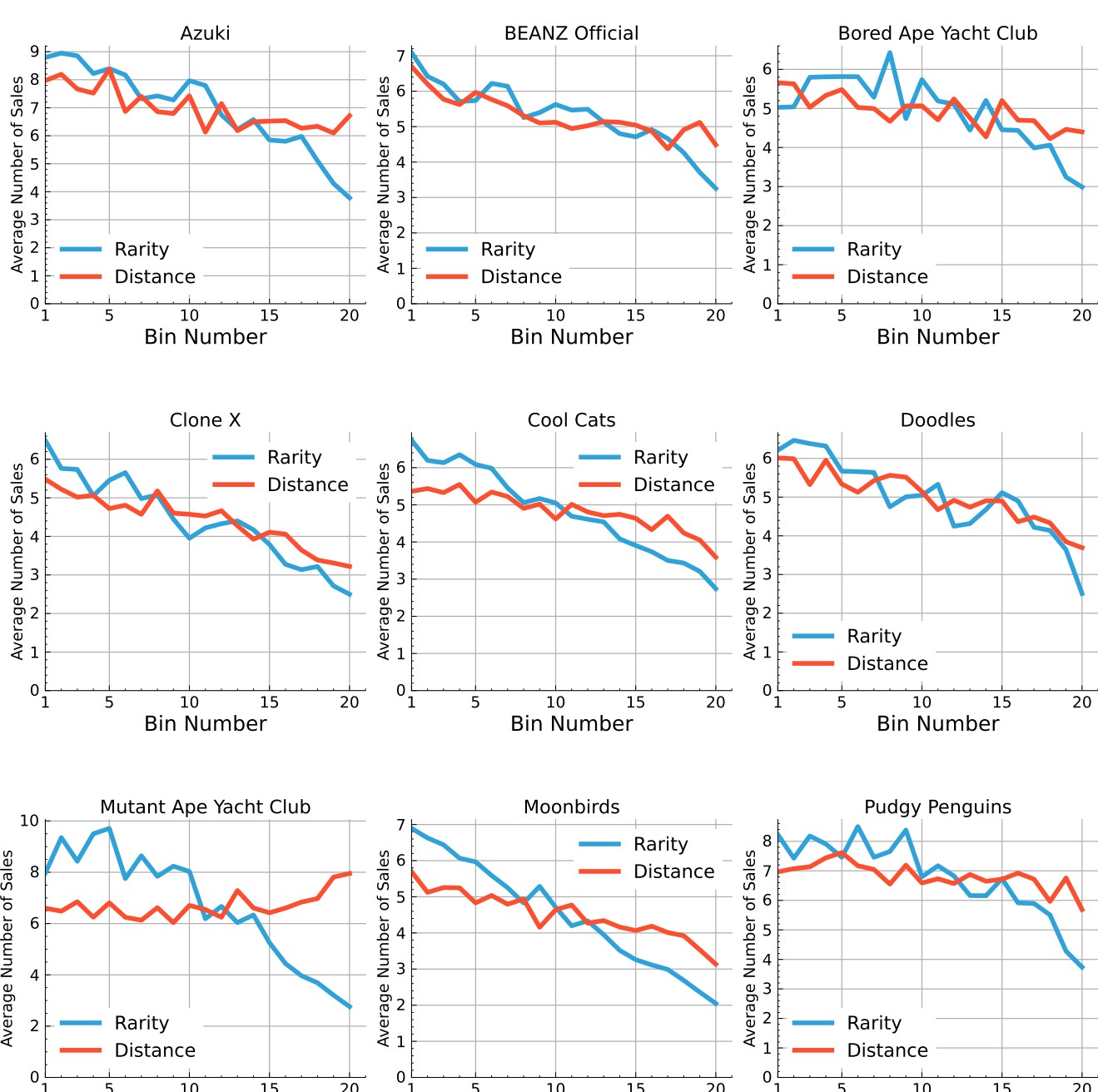

**Figure 7:** *Relationships with number of sales.* **These plots shows the relationship between rarity rank (visual distance) binned into 20 quantiles and number of sales of the NFT. In the case of rarity ranks bins are sorted from highest rarity rank (least rare) to lowest rarity rank (most rare).**

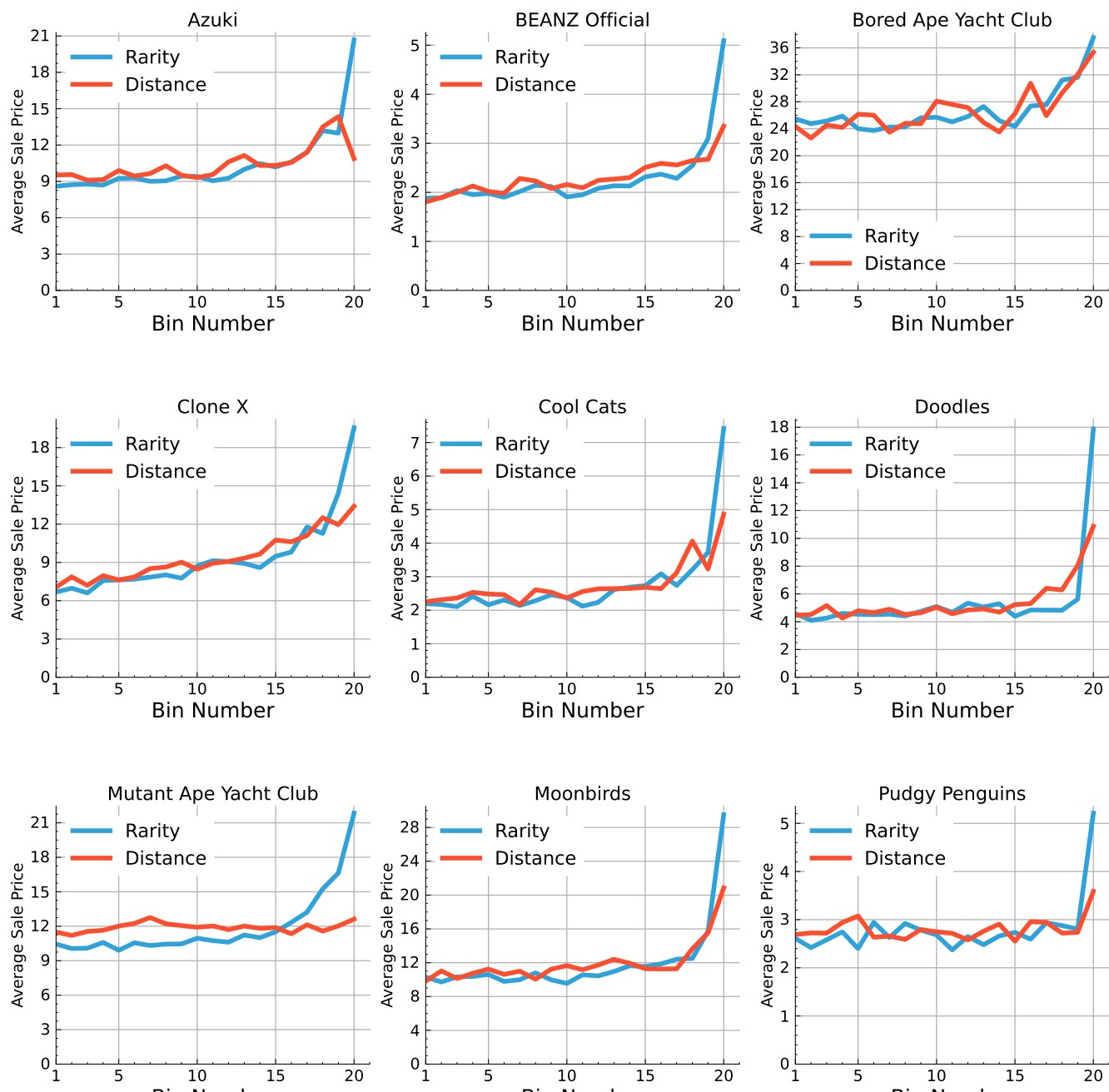

**Figure 8:** *Relationships with number of sales.* **These plots shows the relationship between rarity rank (visual distance) binned into 20 quantiles and average sale price of the NFT. In the case of rarity ranks bins are sorted from highest rarity rank (least rare) to lowest rarity rank (most rare).**

