# OpenReview forum: "NFTs as a Data-Rich Test Bed: Conspicuous Consumption and its Determinants"
_ACM.org/TheWebConf/2025/Conference — WWW 2025 Poster_

### Official Review · Reviewer_5Jn7 · 2024-11-29

**Novelty:** 1
**Technical Quality:** 2

**Review:**

This paper, while ambitious in its attempt to leverage NFTs for studying conspicuous consumption, falls short on multiple fronts. The authors introduce a novel dataset combining NFT transaction data with visual embeddings, but the execution is marred by a lack of methodological rigor. The paper's claim that it provides a rich test bed for understanding the dynamics of conspicuous consumption is undermined by the fact that the analysis seems to be overly reliant on a limited set of NFT collections, which may not be representative of the broader market. Furthermore, the treatment of social meaning, a critical variable in the study of conspicuous goods, is vague and lacks a concrete operationalization, making it difficult to assess the validity of the findings. The use of advanced machine learning techniques, such as vision transformers, is commendable, yet the paper fails to provide sufficient technical details, leaving the reader in the dark about how these models were fine-tuned or what specific parameters were used, thus raising questions about reproducibility. The discussion of the bandwagon and snob effects, while theoretically sound, is presented without adequate consideration of other potential confounding factors, such as broader economic conditions or market sentiment, which could significantly impact the observed correlations. Additionally, the graphical representation of the data, particularly in figures like Figure 6, is poorly explained, adding to the overall confusion. The paper also does not sufficiently explore the unique characteristics of NFTs compared to traditional luxury goods, missing an opportunity to contribute to the broader discourse on digital versus physical assets.

**Questions:**

- The concept of social meaning is central to your analysis, yet it remains somewhat vague. How exactly did you operationalize this variable? What criteria or metrics were used to quantify the social significance of NFTs, and how robust are these measures?

- Your analysis of the bandwagon and snob effects does not seem to control for other potential factors such as broader economic conditions, market sentiment, or external events. How do you account for these variables, and what impact might they have on your findings? Can you provide a sensitivity analysis to address this?

- The study focuses on a limited set of NFT collections. How confident are you that the results can be generalized to the broader NFT market? Are there plans to expand the dataset to include a wider range of NFTs, and if so, how would this affect your conclusions?

- Some of the figures, particularly Figure 6, are not well explained. Could you provide a more detailed explanation of the data represented in these figures and the insights they are intended to convey? Additionally, are there any alternative visualizations that might better communicate the key points?

- The paper does not extensively discuss the broader economic and market context in which NFTs operate. How do you see the current and future state of the NFT market, and how might changes in this market (e.g., regulatory, technological) affect the patterns of conspicuous consumption you have observed?

- Given the novel nature of the dataset and the methods used, have you considered making the dataset and code publicly available for replication and validation by other researchers? If not, what are the barriers to doing so, and how might these be overcome?

-

**Reviewer Confidence:**

3: The reviewer is confident but not certain that the evaluation is correct

**Scope:**

3: The work is somewhat relevant to the Web and to the track, and is of narrow interest to a sub-community

---

### Official Review · Reviewer_w5gd · 2024-11-30

**Novelty:** 3
**Technical Quality:** 5

**Review:**

Quality:
This work explores whether NFTs can be classified as conspicuous consumption and verifies through experiments that NFTs exhibit characteristics of the bandwagon effect and the snob effect, thereby proving that NFTs are a form of conspicuous consumption in the secondary market. The experimental process is rich and the results are clear.

Clarity:
The necessity of some experimental methods is not well-argued. For example, the necessity and reliability of using a trained GNN to represent the NFT market and using this network as the basis for experiments are not sufficiently justified.

Originality:
The research focuses on verifying the consumer goods attributes of NFTs in the market, which has a certain degree of originality.

Significance:
The hypothesis that NFTs belong to conspicuous consumption is relatively conventional, and these properties are foreseeable. The significance of the research is relatively not enough.

**Questions:**

1. Why use a trained GNN to represent the NFT market and conduct control experiments based on this network, rather than using a more interpretable data analysis approach for the NFT market?

2. Can the reliability of using visual features to predict the rarity of NFTs be improved? By modifying or optimizing the process of calculating the centroid-Euclidean distance, or by combining RarityRanks to build a more credible overall prediction paradigm?

**Reviewer Confidence:**

3: The reviewer is confident but not certain that the evaluation is correct

**Scope:**

3: The work is somewhat relevant to the Web and to the track, and is of narrow interest to a sub-community

---

### Official Review · Reviewer_Nv3B · 2024-11-30

**Novelty:** 5
**Technical Quality:** 6

**Review:**

# Overview

This paper provides a thorough analysis of Non-Fungible Token (NFT) data, exploring its potential as a testbed for studying conspicuous consumption. The study aims to highlight parallels between the dynamics of the emerging NFT market for digital goods and the historical luxury fashion market, both of which exhibit conspicuous consumption behaviors. The dataset introduced is extensive and representative, enabling robust analysis. By leveraging embeddings, the authors identify evidence of the two primary effects of conspicuous consumption: the bandwagon effect and the snob effect. Their deep understanding of NFT market dynamics informs the design of the metrics and statistical methods used, effectively guiding the analysis.
The paper’s thesis aligns with the broader trend of growing consumption of digital goods. While much of the existing literature focuses on the transition of physical luxury goods from established brands into the digital realm via NFTs, this study uniquely examines NFTs as a standalone asset class.

# Strengths (+) and Weaknesses (-)

(+) Strengths:
- The authors demonstrate a strong understanding of the NFT domain, which is evident in the tailored metrics and statistical methods employed (e.g., wealth and affinity computations, use of Graph Neural Networks, rarity computation). These approaches reflect the specific dynamics of the NFT market.
- The assumptions and explanations provided to justify the results are insightful, offering valuable perspectives on NFT value creation, community building, and consumer behavior within this emerging field.
- The paper is well-illustrated, with clear figures showcasing example NFTs to clarify metric definitions and graphs to present results. Notably, Figures 2 and 5 are particularly descriptive and self-explanatory.

(-) Weaknesses:
1. Minor oversights:
- Clarify the term "information content" (line 265) in the context of rarity ranking.
- Overlay an "Ideal Line of Fit" on the "Line of Best Fit" to improve visual assessment of performance.
- Potentially add the word "token" in the phrase "sale price data for every X allowing" (line 723).
2. Figure 3 is challenging to interpret despite the caption. While it provides a structured view of the results, a simpler visualization, such as a histogram or heatmap, could enhance clarity.
3. The study could benefit from incorporating Natural Language Processing (NLP) analysis on social media platforms like Twitter and Reddit to link public sentiment about NFT collections with their conspicuous consumption dynamics.

**Questions:**

1. Today, NFTs are not as popular as they were when the technology was launched in 2021. How could this observation be highlighted in contrast to physical luxury goods (bags, watches...)?
2. To which extent could we quantify the relationship between marketing of a certain collection of NFTs, and its perceived rarity or sales volumes ?

**Reviewer Confidence:**

3: The reviewer is confident but not certain that the evaluation is correct

**Scope:**

4: The work is relevant to the Web and to the track, and is of broad interest to the community

---

### Official Review · Reviewer_xUu5 · 2024-12-02

**Novelty:** 3
**Technical Quality:** 6

**Review:**

Strengths: 1. the paper provides a novel dataset consisting of NFTs in their secondary resale market that could potentially enable better analysis of conspicuous goods.
2. the paper provides detailed technical analysis of the bandwagon and snob effects in the NFT market. In particular, the paper does a good job in addressing that simply showing the predictive power of visual distinctiveness in predicting prices is not enough, and we need more than just correlation.
3. The paper is well-written and easy to follow.

Weaknesses: 1. while the NFT market does exhibit two of the properties of conspicuous goods, it is not clear how well the findings from this dataset would generalize to a broader context.
2. the paper could also benefit from additional research question(s) that demonstrate the applicability of this dataset. For example, are there research questions that are difficult or impossible to answer using theoretical models but feasible to answer using this dataset? This would help solidify the contribution of the dataset beyond theoretical models.

**Questions:**

1. See weakness #2 above.

2. Is demonstrating the bandwagon and snob effects enough to justify that NFTs are conspicuous goods? I assume it is from the description of the authors' contributions, but it would be better to clarify this in the paper.

3. Have the authors considered the effects of community diffusion (e.g. through online social networks, through discord etc.) on the value of NFTs?

**Reviewer Confidence:**

3: The reviewer is confident but not certain that the evaluation is correct

**Scope:**

3: The work is somewhat relevant to the Web and to the track, and is of narrow interest to a sub-community

---

### Official Review · Reviewer_vTgm · 2024-12-03

**Novelty:** 2
**Technical Quality:** 4

**Review:**

Pros:
1.	Technical Robustness: The authors use Graph Neural Networks (GNNs) to effectively capture complex relationships in the ownership graph, integrating both structural and visual information. This robust approach enhances the model's predictive power.
2.	Large-Scale Data: The study leverages a large and rich dataset, including nearly 50 million wallet-token pairs and over 370,000 unique wallets, demonstrating the authors' capability to handle and analyze extensive, complex data.

Cons:
1.	The idea that NFTs are part of conspicuous consumption, signaling wealth and taste, is not original. This concept is well-established in the literature on luxury goods and social signaling, particularly through Veblen’s theory of conspicuous consumption. NFTs, as a modern form of digital luxury goods, fit neatly within these existing frameworks, making the theoretical contribution feel predictable rather than innovative.
2.	The paper does not offer a novel theoretical framework or deep insights into the relationship between consumption, signaling, and social networks in NFTs. The notion that NFTs signal wealth and taste is almost self-evident in the context of luxury markets, and the contribution to existing consumption theory is minimal.
3.	The focus on conspicuous consumption is narrow and would benefit from incorporating other aspects of NFT value, such as aesthetics, community, or scarcity

**Questions:**

1.	Although the use of GNNs is justified for this task, could the authors discuss why they chose GNNs over other state-of-the-art graph-based models, such as GATs or R-GCNs, which might also be applicable?

2.	Could the authors provide a comparison with a broader range of models to further validate the effectiveness of their GNN approach? The current comparison seems limited, and including more state-of-the-art models would strengthen the robustness of the results.

**Reviewer Confidence:**

3: The reviewer is confident but not certain that the evaluation is correct

**Scope:**

3: The work is somewhat relevant to the Web and to the track, and is of narrow interest to a sub-community